# From Crisis to Opportunity: Developing a Virtual Marketplace to Enhance Sustainability and Resilience in Small-Scale Fisheries

Luca Bolognini [1,*], Cristina Frittelloni [2], Francesca Perretta [2], Martina Scanu [1,3] and Fabio Grati [1]

1    CNR IRBIM, National Research Council—Institute of Marine Biological Resources and Biotechnologies, Largo Fiera della Pesca, 60125 Ancona, Italy
2    Marche Agricoltura Pesca—Agency for Innovation in the Agri-Food and Fisheries Sector, Via dell'Industria 1, 60027 Osimo, Italy
3    Department of Biological, Geological, and Environmental Sciences (BiGeA), Alma Mater Studiorum—University di Bologna, Piazza di Porta S. Donato 1, 40126 Bologna, Italy
*    Correspondence: luca.bolognini@cnr.it

**Abstract:** In a context in which climate change, overexploitation, and environmental degradation are continuously progressing, the sustainable use of the sea is a key target, both for resources and fishery operators. With the aim of enhancing the sustainability and competitiveness of the entire fishery sector, an innovation brokering event was organized. Via the application of a participatory, interactive, and bottom-up approach, different actors in the Italian SSF sector were stimulated to work together to find innovative solutions to day-by-day problems. During the consultation, and between the identified problems, emerged the lack of cooperation for marketing activity and effective strategies for seafood product development. A pilot action consisting in co-designing a virtual marketplace (VirMa) was the result of the event, intending to facilitate the direct sale of seafood products, explore this new and valuable market, and establish a direct line with consumers. The VirMa application was developed as a value chain network, trying to add value to seafood products. However, an information and communication tool, such as a VirMa, could contribute to increasing the resilience of the SSF sector to market constraints even during unforeseen events, such as a pandemic.

**Keywords:** small-scale fisheries; value chain; information and communication technology; co-design; virtual marketplace

**Key Contribution:** Information and Communication Technology (ICT) can effectively contribute to supporting the value chain of the small-scale fisheries (SSF) sector (technology in the artisanal sector). Co-design is a valuable way to involve different SSF actors all together. Tools such as a marketplace, as well as labeling, traceability, e-commerce, seafood transformation, and seafood delivery, must be considered as opportunities for SSFs. The SSF sector must be ready for unexpected events (a lesson learned from the COVID-19 outbreak). Environmental, economic, and societal sustainability are evermore essential objectives in the SSF sector.



## 1. Introduction

Fishing has always been an activity of vital importance not only because it can provide food with a high nutritional value [1,2] but also because it contributes to human well-being with induced benefits, including economic and cultural ones [3]. However, one of the greatest challenges facing modern societies is to try to provide food and livelihoods for a rapidly growing population, in a context in which the effects of climate change and environmental degradation are having a strict effect on resources. Intending to achieve a better and more sustainable future for all, the United Nations in 2015 set sustainable development goals (SDG) to be implemented by the year 2030. In particular, SDG 17, "Strengthen the means

of implementation and revitalize the Global Partnership for Sustainable Development", focuses on a vision for improved and more equitable trades. It represents a unique, transformative, and integrative approach towards the sustainable and resilient path that leads to a planet that leaves no one behind [4,5].

The Mediterranean fishing sector shows a marked heterogeneity in terms of exploited species, fleet composition, and socio-economic and political complexity. The dominant component of the fleet, in terms of the number of vessels, is represented by the artisanal fishing segment, made up of multipurpose boats less than 12 m long and with no towed gear [6], which represents about 80% of the total fleet and the largest number of employees [7]. This sector not only draws livelihoods from the oceans but also connects land and sea domains, with a social and cultural role for the communities [8]. At the same time, it contributes to providing fresh fish to the local and international markets [9], representing an important asset not only as a food supply but also for local economies, especially in fishing-dependent areas [10].

In the context of the overexploitation of biological resources, the key role played by the small-scale fisheries (SSF) sector was recognized worldwide, especially in the achievement of major sustainability targets [11–13]. This sector, compared to the large-scale fishing fleet, has a crucial role in poverty alleviation and eradication, in addition to food nutrition and the sustainable exploitation of marine resources. This activity is one of the most relevant in the fisheries sector, notably among coastal communities, both in terms of employees and catches (representing 90% of the world's capture fishers and fish workers). The Food and Agriculture Organization of the United Nations (FAO) has even identified the SSF sector as one of the main actors being fundamental in achieving targets such as food security (SDG1), reduced poverty (SDG2), community well-being (SDG3), gender equality (SDG5), and economic growth (SDG8) [14].

At this moment, a series of important and promising initiatives and opportunities are in place to pursue ambitious objectives and may contribute to the prosperous future of the sector; supporting poverty eradication; enhancing the progressive realization of the right to adequate food [2]; promoting at once a sustainable future for the planet under an economic, social, and environmental point of view; improving the socio-economic situation of fish workers; and providing guidance for state and stakeholders for the participatory policies that are ecosystem friendly [15]. In this view, the value chain of seafood products has increased interest and investments toward the valorization of fishery production (e.g., eco-labeling, quality trademark, etc.). Looking at the SSF sector, its value chain could also reveal response strategies that enhance the sustainability and competitiveness of the entire sector and the operators involved in it [16].

Nevertheless, the vulnerability to the global market system of the SSF sector was recently revealed as one effect of COVID-19, immediately after its occurrence [17]. A global pandemic might be once in a lifetime; however, other natural phenomena, such as disasters, or anthropogenic phenomena, such as recession, political instability, and trade wars, are common and could lead to an unpredictable market shock. For those reasons, an international solution is crucial to protect against global market volatility.

A potential solution will be the development of insurance opportunities to safeguard SSFs, so it may avoid future crashes, or a market diversification, which is recognized by the FAO as a key to SSF sustainability that could force it to face such events. Other key aspects for increasing SSF resilience are represented by creating new and premium demand, including designing and implementing market diversification [17].

To this end, different studies investigating the generation of ideas for technology-based services found that involving ordinary users at the front end of the development process resulted in the creation of new ideas and products [18,19]. The present work aims to report and analyze results coming from a local co-design exercise to convert the specific needs of SSF stakeholders (fishers and buyers) into a concrete information and communication tool (ICT), such as a virtual marketplace, to pinpoint the requirements of the value-chain actors.

## 2. Materials and Methods

In the framework of a cooperation initiative (funded within the ARIEL Project—Interreg VB ADRION program) held in the Marche Region (Italy), an innovation brokering event was organized in February 2019 to promote a cross-fertilization activity among stakeholders, enterprises, and researchers involved in the SSF and aquaculture (AQ) sectors. This activity was made to stimulate actors from different backgrounds to partner and cooperate around core issues in day-by-day operations and to put into practice technological and non-technological innovation.

The process consisted in the application of a participatory, interactive, and bottom-up approach to stimulate different actors of the SSF and AQ sectors (operators, advisors, academics, researchers, and policymakers) grouped in working groups (WGs) to discuss core topics and find potential solutions to stimulate innovation.

Actors were grouped according to their specific interest/influence on a topic and were facilitated by moderators towards the innovation discovery process in highlighting their day-by-day needs to find quick and fit-for-purpose technological or non-technological solutions to put into practice.

The innovation brokering event was organized by applying the Open Space Technology methodology [20] and was structured in several steps:

- Participants' invitation (weeks before the event);
- Participants' registration and categorization (at the very beginning of the event);
- Plenary session;
- Working group discussion;
- Instant reports from each working group;
- Discussion in plenary.

Findings from each group were presented and separated into several bullets as listed below:

- Main problems listed according to importance;
- Proposals for solving problems;
- Possible solutions;
- Need for specific innovations;
- Pilot studies;
- Probability of implementation.

After the presentation of the findings of each group, there was a brief discussion with stakeholders to assess the feasibility of the proposed activities. Moderators proposed a framework agreement on the implementation of the selected activities, incorporating them into the project. After listing and describing all proposals from all WGs, facilitators invited the stakeholders to give their feedback and opinions on the presented proposals of each of the groups.

For the specific pilot testing, additional stakeholders from the different thematic areas were invited to contribute to the exercise. Based on inputs and feedback provided during the pilot action activities by the involved WG members, the research team cooperated with the software development team (SDT), with the scope of transforming the conceptual workflow into an Information and Communication Technology (ICT) application.

## 3. Results

### 3.1. Consultation

During the stakeholder consultation activities, a total of 70 participants agreed to be involved in the following phases. A total of 44 stakeholders participated in the innovation brokering event (Table 1). After a first brief discussion, six specific thematic tables were arranged to locate homogeneous respondents. Stakeholders were divided into four groups according to previously determined core topics, setting-up regional SSF and aquaculture cross-sectoral working groups (WGs): (1) Process Innovation; (2) Product Innovation; (3) Energy efficiency in AQ and SSFs; and (4) Traceability, Marketing, and Branding Seafood

Products. This configuration allowed the stakeholders to perform and discuss topics in depth, providing strategic direction. A total of 10 identified stakeholders participated in the table related to the SSF "Innovation market-oriented" topic: one local action group representative, four operators, and five business entrepreneurs (Table 1). Based on the above-mentioned consultation, the WG participants expressed their specific needs: the development of a cooperation network among operators (50% preferences), the valorization of seafood products (30% preferences), and the allocation of economic resources for the establishment of a solidarity fund (20% preferences). These requirements were expressed mainly by the AQ sector. In accordance with the participants, the most relevant problem affecting the sector was tackled for subsequent pilot action. The main issues that arose from the market aspect were identified: competition, as one of the most relevant causes of price depletion, and the absence of infrastructure, such as logistic or market infrastructure. Additional general aspects, such as the lack of cooperation for marketing activities, and an effective strategy for seafood product development were recognized as reducing the possibility for the SSF sector to penetrate the main and most valuable market segment. Moreover, other constraints to the development of SSFs were identified as attributable to the intrinsic characteristics of their companies, which are comparable to mini- or micro-enterprise dimensions, that inevitably lead to the absence of structure/organization, forcing the fisher to face up to several day-by-day duties, including fishing operations, boat and gears maintenance, and, above all, administrative procedures. Additionally, very often, the sector suffers from competition with other fleet segments, such as bottom trawling, that may cause an increase in the prices of fishing tools, massive landings that may depress fish prices [21], or the conditions for a monopolistic market that inevitably leads to the economic unsustainability of the sector (Table 2).

**Table 1.** List of stakeholders attending the innovation brokering events; the innovation-market-oriented working group, specifically; and the focus group involved in pilot testing.

| Stakeholders | Participants | | |
|---|---|---|---|
| | Innovation Brokering Event | Working Group | Focus Group |
| Policy makers | 1 | 1 | |
| Small-scale fisheries local action group | 1 | 1 | |
| Education, research, and academia | 20 | 3 | 5 |
| NGO | 1 | 1 | |
| Aquaculture operators | 4 | 2 | |
| Small-scale fisheries operators | 4 | | 4 |
| Recreational Fisheries Association | | 1 | |
| Ichthyotourism operator | | | 1 |
| Restaurants | | | 3 |
| Solidarity buying groups | | | 2 |
| Regional development agency | | | 1 |
| Certification and branding experts | 5 | 1 | |
| Small-scale fisheries and aquaculture Entrepreneurs (business) | 5 + 2 | | |
| Shipbuilding | 1 | | |

With the aim of overcoming the specific market constraints described above, all the stakeholders agreed to undertake a pilot action related to the co-design of a virtual marketplace to facilitate the sale of seafood products coming from the the SSF sector, exploring the new and most valuable market and establishing a direct line with consumers. Then, additional stakeholders effectively contributed to the design of the virtual marketplace application, according to specific needs related to their area of interest. In particular, this focus group was composed of five members from the educational, research, or academic sectors; four SSF operators; one fishing tourism operator; three restaurants owners; and two

solidarity purchasing groups (SPGs) (Table 1). Each group worked together with the SDT. The work of this focus group started immediately after the innovation brokering event and continued until December 2020, dealing with fishery stakeholders and the consequences of the COVID-19 pandemic on the value-chain.

**Table 2.** List of stakeholder's issues and needs grouped by sector (bullet points represent each stakeholder group: ● SSF operators; ○ buyers (all); □ solidarity purchasing groups; § research; * read the discussion for details).

| SSF Issues | | Common Needs | | Buyers Issues |
|---|---|---|---|---|
| | ● | New and valuable market | | |
| | ● | Free prices policy | | |
| | ● | Affordable smart technologies | | |
| | ● | "Portability" of application | | |
| ● Market competition with other fleet segments (almost monopolistic) | ● | Low click input | □ | Buy fresh fish food * |
| | ● | Real time management of the market | □ | Lack of fresh seafood products ready to eat |
| ● Price depletion | ● | Easy management of the orders | ○ | Home/restaurant delivery |
| ● Absence of market infrastructures | | | ○ | Difficulty in finding some specific types of seafood |
| ● Lack of cooperation for marketing activities | ○ | To be informed on the origin of products -> traceability | ○ | Lack of info on the market availability |
| ● Lack of strategy for the seafood product development | ○ | To be informed on market availability | ○ | Knowledge on the origin of products |
| ● Mini- or micro-enterprise dimensions | ○ | To establish a network of trust (more info on fisher, boat, and gear uses) | ○ | Know the entire supply chain ("storied seafood" *) |
| ● Absence of structure/organization (e.g., co-op) | ○ | To make responsible purchases | ○ | Lack of trust in the seller |
| ● Day-by-day duties | ○ | To eat the right kind of seafood products | | |
| | § | Visualization of the price fork (minimum and optimal species price) | | |

*3.2. Co-Design of a Virtual Marketplace (VirMa)*

In the above-mentioned context, an ad hoc focus group was created to perform a co-designed exercise to identify the best innovative approach to overcome the previously emerged issues. The pilot action activities started before the COVID-19 crisis outbreak, but the work continued during the pandemic restriction. The fact that the focus group was instituted before the pandemic outbreak, and that the stakeholder network was consolidated prior to the enforcement of social restrictions, was one of the reasons that led the work to be completed.

The results herein reflect the work made following the cyclical workflow: conceptualization, implementation, digitization, and feedback from users. The characteristics of the ICT tool created during this project is the final product of the above-mentioned process and was considered completed when all the users deemed it to be "fully functional".

The VirMa application was developed by the SDT using FileMaker© software (Claris International Inc., version 18; Cupertino, CA, USA). As the first step of the co-designing process, the application, based on the specific profile, was able to run both on smart devices, such as smartphones and tablets, and on personal computers. The "portability" of the application was a prerequisite specifically expressed by SSF operators. Each profile was generated by an administrator, who created the account based on user groups. The user

groups were identified as fishers, restaurants, and private buyers. Each profile had specific privileges. To increase the awareness of buyers (restaurants and private ones), as explicitly asked by their representatives in the WG, specific information related to each fisher's profile was added. Such information was, of course, associated with the fisher's identity, the name and picture of the boat, the boat code, and the registered harbor (Figure 1). During each fishing day, fishers could add a fishing session that was automatically generated, based on the calendar day. As requested, a pre-charged list of species was added to the application to facilitate the fishers' input, decreasing as much as possible the number of clicks needed (Figure 1). For each species, the fisher could insert the gear used, the catch amount, and the price. The price policy was left totally free to mimic the actual market, and each operator could choose the best price fitting with his daily catch; nevertheless, the SSF operators agreed to visualize the minimum and the optimal price suggested for each species. This price fork was established according to periodical consultations with fishers about prices and was expressed as mean value (Figure 1). All the inputs uploaded to the VirMa application by fishers were available in real time to the buyer interface. Indeed, as they requested, buyers could visualize the seafood products sorted by fishers, harbor, or species; for each species, the instant availability was visualized on screen, and then they could place their pre-order (Figure 2). Simultaneously, both fishers' and buyers' accounts could interrogate the VirMa application to obtain detailed information concerning, respectively, the pre-orders received and placed. Moreover, in each daily fishing session, there was a specific box where fishers could add a note or whatever information they wanted to share with buyers (photos, fishing area, etc.). Due to the time availability and the purpose of this exercise, the electronic transactions functionality was not taken into account, essentially due to the effort needed for proper e-wallet development and management. For additional information, see [22].

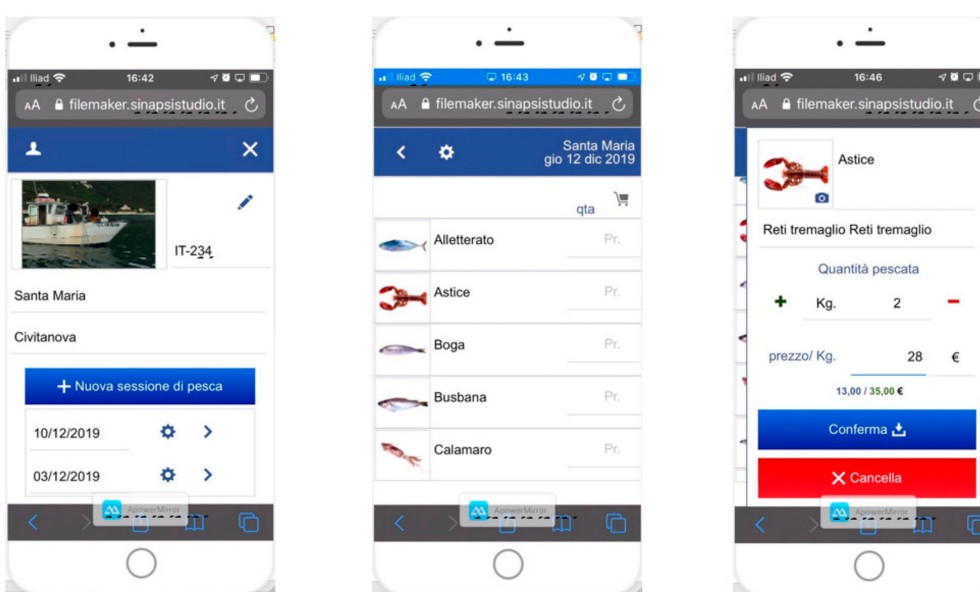

**Figure 1.** Screenshots of the VirMa application from a fisher's account: new session page, species list page, and species amount and price.

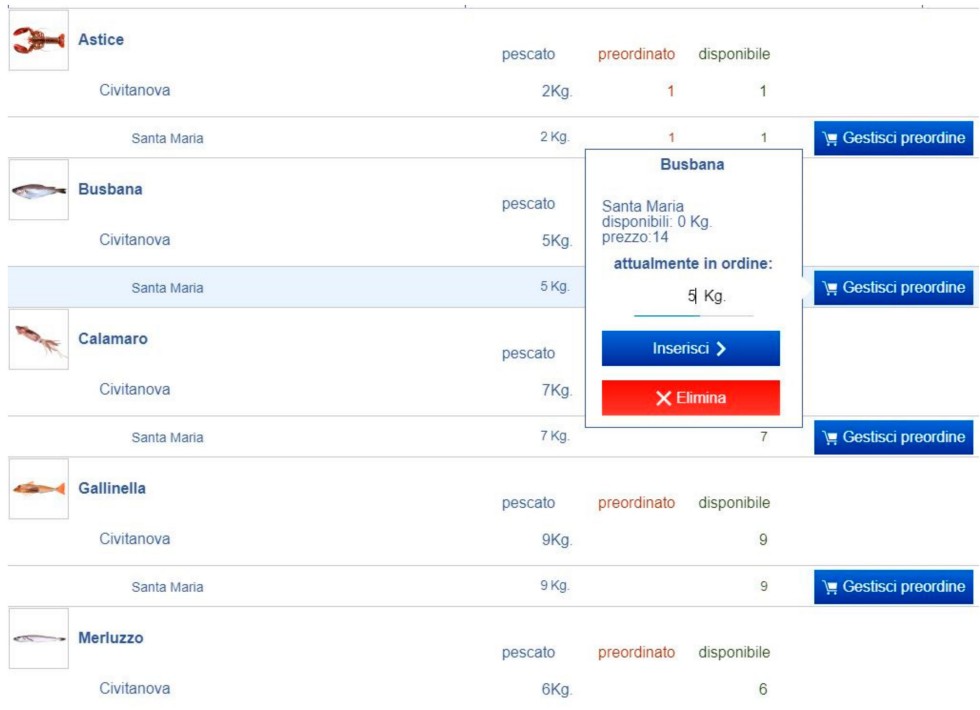

**Figure 2.** Screenshots of the VirMa application from a buyer's account: the list of seafood availability sorted by species, by harbor, and by boat. The pop-up shows the pre-order input.

Unfortunately, due to the restrictions imposed under the COVID-19 pandemic and the project timeline, the pilot testing of the VirMa application was only performed by the stakeholder group involved in its development. For a few weeks, fishers and buyers interacted using the app to show available products and place orders.

## 4. Discussion

During the innovation brokering event, several issues emerged from the local stakeholder consultation with the SSF sector, especially related to the market aspects. This work briefly reports the main gaps that emerged and a possible way to cope with the problem related to the value of SSF products and access to a sustainable market. Here, it is highlighted how a co-design exercise can help in identifying a possible solution to increasing the revenue of a sector, particularly one that is affected by unexpected events, such as the outbreak of coronavirus 2 (SARS-CoV-2).

Based on encouraging local community participation coming from the Abalobi© (Cape Town, South Africa) initiative [23], especially in the context of reversing the devastating effects resulting from COVID-19 for many fishing communities [24], structuring a virtual marketplace in a co-design exercise was identified as a shared solution during the abovementioned consultations. This participatory design allowed all the stakeholders to work together to align their ideas toward a common goal. Even if they had different competencies or diverse operational levels, the activity transformed all participants into co-authors of the application.

The resulting VirMa app was developed as a value chain network, by which SSF products could directly move from the point of production to consumption, minimizing links in the chain and aiming to maximize profits [25]. However, a value chain can be seen as something more than bringing the product to market; its scope is to provide a more mutually beneficial environment for all stakeholders [2], incrementing the final value of the product. In the fishery sector, "value addition" generally includes some type of processing method, but recently, the concept of "value creation" is coming forward. It consists of enhancing product attributes via traceability, environmental stewardship, a

direct relationship with the fisher, etc. Therefore, value chains can be viewed as empowering to the various, usually fragmented, stakeholders as they recognize innovative opportunities to contribute and increase their product value. All over the world, many initiatives with this view are arising: E-Fish (Infoteam S.r.l; Pescara, Italy), FishLine® Fresh Local Seafood (Phondini Partners; Half Moon Bay, United States), YORSO Fresh Fish B2B (Yorso Group. Inc; Tallinn, Estonia), and Fresh Fish Alert (Dipartimento Pesca Mediterranea Regione Sicilia and Università degli Studi di Catania; Catania, Italy). Despite that, the authors consider the co-design approach as the strength of the VirMa application compared to similar products.

A co-design approach is considered to be more original and creative compared to other traditional approaches [26–28], and it has, in general, been reported to improve the efficiency and effectiveness of its products [29]. The direct users' input integrated with the designers' ideas via co-creation improves the efficiency of a product facilitating continuous product/service improvements and reducing the risks of failure; on the other hand, the product developed by this approach will better match customer's needs, increasing its effectiveness [18]. Trott's studies [19] investigating the generation of ideas for technology-based communication services found that involving ordinary users at the front-end of the design process resulted in the generation of both radical and incremental ideas for new products. Direct users' engagement was the key in developing the VirMa application, as opposed to simply listening to their opinions, which could have led to misinterpretation [30].

A fundamental criterion for fisheries management, followed during VirMa development, was that the fisheries and associated value chains were structured in a way that they did not incentivize overexploitation, as suggested by the FAO (1995) [31]. Fundamental points in designing the application were (1) the minimization of the supply chain facilitating the direct sale to restaurants or direct buyers, allowing fishers to earn more money by deleting links in the chain; (2) the enhancement of product value via traceability information; (3) the promotion of the SSF sector as a low-impact activity, underlining the seasonality of the product and sharing the high selectivity of the gears; and (4) the promotion of the more sustainable exploitation of the sea, proportioning catches according to demand. From the consumer's point of view, information on fishers, boats, gears, etc., are essential to recognizing what is behind what they are eating. This is also one of the most relevant models promoted by Abalobi©, using the concept of "storied seafood". If consumers rethink the way they eat fish, they will be clever to recognize the value of a more sustainable and ethical food system, thanks to seafood with a social and ecological story [32]. Our economies and societies, and people's lives, have been transformed by digital and other technologies [33]. There is literature about the emerging evidence on the benefits of Information and Communication Technology (ICT) as a useful tool for fisheries. ICT can increase the socio-economic level, knowledge and skills, and communication processes and enhance the safety aspects of fishers [34], offering a variety of tools for cataloging (e.g., data collection), analysis, and validation, in addition to connecting separate actors [35]. One of the most relevant features of ICT is to make use of it as a lever for value chain upgrading and influencing SSF contributions to marine resource sustainability [36].

The results of the co-design exercise here are presented to demonstrate the central role played not only by fishers but also by all the actors involved in the SSF sector, highlighting the importance to align specific needs arising from each target group to the whole value chain. The engagement of actors in future ICT seminars or workshops is desirable [34] because it could contribute to facilitating the democracy and socio-economic reform of SSFs [36] and potentially bridge the gap between fishers and research [37], avoiding the consideration of fishers as passive generators of data [36].

## 5. Conclusions

The co-design process and the user experience included in this work were considered the prerequisite for successful idea and product generation, addressing the direct experience of the problem that actual or potential users could develop. Burns and colleagues [38]



discussed this approach as a way to manage change processes and to promote creativity and innovation so that the people involved can engage in continuous learning and innovating.

The adoption of a new ICT tool in the fisheries sector in general will be affected by several factors, such as costs, ease to use, time, etc., and the debate on which ICT tool is more beneficial than others is a debate still ongoing among researchers [39]. However, the results from this case report and other works are encouraging, suggesting the potential capability of ICT tools to effectively contribute to promoting economic, social, and environmental targets for SSF. Moreover, in a pandemic context, in which social distancing is affecting traditional markets, an ITC tool such as the VirMa application can play a role in reconnecting sellers and buyers.

Nevertheless, the growing and uncontrolled development of such kinds of tools could undermine the principle for which it has been used until now. Based on that concern, the authors strongly encourage the adoption of a clear and transparent policy vision to define the fundamental principles essential to pursuing social, economic, and environmental sustainability in the whole SSF sector.

**Author Contributions:** Conceptualization, L.B.; methodology, L.B.; investigation, L.B., C.F., F.P., M.S. and F.G.; writing—original draft preparation, L.B.; review and editing, L.B., M.S. and F.G. All authors have read and agreed to the published version of the manuscript.

**Funding:** This research was funded by the Adriatic-Ionian Programme INTERREG V-B Transnational 2014–2020 First Call for Proposal, project number 278.

**Institutional Review Board Statement:** Not applicable.

**Informed Consent Statement:** Informed consent was obtained from all subjects involved in the study.

**Data Availability Statement:** The data presented in this study are available in the manuscript.

**Acknowledgments:** M.S. contributed to the research leading to these results while enrolled in the Ph.D. Program "Innovative Technologies and Sustainable Use of Mediterranean Sea Fishery and Biological Resources-FishMed". The authors are also thankful to all the stakeholders that cordially contributed to this work, without which its realization would not have been possible.

**Conflicts of Interest:** The authors declare no conflict of interest.

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
