# Peer review of "From Crisis to Opportunity: Developing a Virtual Marketplace to Enhance Sustainability and Resilience in Small-Scale Fisheries"

_fishes, doi:10.3390/fishes8050272_

Round 1
Reviewer 1 Report
the article is well written but it has gramatical and phrases related mistakes those need to be rectified before it can be accepted for publication
1. What is the main question addressed by the research?
Reply: the topic it self is very emerging as UN is looking towards SDGs and small scale fisheries is one of the growing and important sector to achieve this goal in years to come.
2. Do you consider the topic original or relevant in the field? Does it
address a specific gap in the field?
Reply : it’s very interesting and demanding topic for the policy makers.
3. What does it add to the subject area compared with other published
material?
Reply : it is an addition to the existing knowledge on small scale fisheries and its expansion and effective management through virtual marketing
4. What specific improvements should the authors consider regarding the
methodology? What further controls should be considered?
Reply: its al right
5. Are the conclusions consistent with the evidence and arguments presented
and do they address the main question posed?
Reply: yes
6. Are the references appropriate?
Reply: Yes, but I suggest author to provide recent references in introduction and discussion section.

it can be accepted for publication with minor revision in terms of languages, gramatical mistakes.
Author Response
The authors wish to thank you the reviewers for the suggestion provided; here below is the reply:
The article has been revised from a grammatical point of view.Reviewer 2 Report
The authors present a paper looking at the co-design of a sales/marketing app for SSFs. I think this is really interesting and worthy of publication but at the moment the manuscript lacks focus or a strong narrative and it is hard to follow at times. One of the main interesting aspects of this work is the co-design approach but there is limited detail of how the app was collectively developed. At the moment the paper gets distracted by the mention of Covid. While this obviously has a big impact on fisheries and the app may be useful in mitigating against some of these it wasn't initially designed to deal with problems as a result of Covid and again this causes the paper to lose focus. I think some restructuring is required to make this more focussed.
The introduction would benefit from some restructuring. Currently the introduction talks about SSF but again not in any structured way. The authors briefly introduce the Mediterranean including some issues about overfishing, although only 1 species is specifcally mentioned here. Then the authors go on to make much more generalised statements about SSFs that seem more relevant to smaller, subsistence fisheries. It could be worth starting by describing SSFs as a whole and how these vary in different regions before focussing in on the Mediterranean. Currently it jumps around and can be hard to follow the point that is trying to be made and exactly what problem the work is addressing. There is no real introduction of the concept of co-design approaches and their benefit in SSFs either, this would be a useful addition to the introduction.
The overall objectives of the workshop need to be made clearer. There is a lot of mention of Covid in the manuscript, but I don't think this was the main driver of the work, there needs to be a much clearer explanation as to why the coproduction workshop was organised and what was hoped to be achieved from it.
At times it feels like Covid is being shoehorned into the manuscript. The app was originally developed prior to Covid due to other issues the industry felt were important. While Covid has been another stressor on the industry there is no evidence in the paper of how the app has been useful since the pandemic (i.e. no relfections from stakeholders or anything to say how it has been used to mitigate against effects of Covid). While Covid has obviously been an issue for many fisheries I'm not sure how relevant it is to this application. This either needs to be better described or the paper needs to be restructured so that there is better balance between all of the issues of concern for the industry and how the app addresses this.
The discussion again focusses on Covid, but I think it would be better to focus on the co-production process and the other key outputs from the workshop (i.e. the things the participants identified as being key concerns and how the app developed might help to address these). I'm not sure how operational the app is or if it has been tested or used by industry but some reflection on this, it's uptake/success could be beneficial here.

In the uploaded version of the manuscript I have highlighted a few sentences that are unclear and added a few suggestions to changes/deletions from the text that would make it a little easier to follow.
Author Response
The authors wish to thank you the reviewers for the suggestion provided; here below is the reply:
The Covid parts were reduced. The introduction, as well as the discussion section, was revised.

Reviewer 3 Report
Overall, the concept of the paper is interesting (proposing an ICT tool aimed at supporting SSF), yet the presentation of the material is rather poor.
The Introduction part is quite general and it contains many redundant and irrelevant paragraphs.
The methodology is insufficiently explained, both when referring to the stakeholder consultation (the co-design brokering event) and the actual design of the VirMa application.
The Results and Discussion sections tend to be a little more insightful. However, they need to be additionally developed, in order to meet the requirements of a research article, and not just a short communication.
I recommend reconsidering this manuscript for publication only after a major revision is made, both as content and English proofing. Extensive editing of English language required.
Additionally, the following specific comments below must be addressed:
Specific comments
Lines 13-14 (Abstract): Avoid using “overexploitation“ and “exploitation“ in the same sentence.
Line 16 (Abstract): Grammar correction needed (a gerund is required) - “with the aim of enhancing“.
Lines 21- 22 (Abstract): Grammar correction needed, a gerund is required here as well - “A pilot action consisting in co-designing... “.
Line 23 (Abstract): Grammar correction needed (gerund required) - “...and establishing a direct... “.
Lines 44-48: Rephrase the entire sentence on UN’s Sustainable Development Goal 17, as it is unclear and confusing for the reader.
Line 64: Erroneous construction of the the sentence, there is a redundant use of “it“, which repeats the subject (“specific stock“). Reconstruct the entire sentence.
Line 88: Wrong past participle used here: the needed verb is “to arise“ (not “to arouse“, with a totally different meaning!), with the correct phrasing “the value chain of seafood products has arisen“.
Line 91: Wrong word order here: “operators involved in it“ is the correct version.
Line 102: What does “bring to“ mean here? Please clarify!
Line 125-126: Wrong use of the gerund here, correct “to partnering“ to “to partner“.
Line 162: Be consistent when writing Information and Communication Technologies (ICT) - with capital letters everywhere in the manuscript.
Line 167? Unusual formulation, replace “Italian country“ with simply “Italy“.
Line 168, Line 176, Line 204: Be consisted with table citations in the text: either Tab. X or Table X. Also, choose and maintain either Roman or Arabic numerals throughout the whole manuscript.
Line 173: Wrong word, replace “deep“ with “depth“.
Table 201: Replace “accordingly with“ with “according to“.
Line 207: Replace “attended“ with “attending“.
Line 220: “What here reported“ is a simplistic and erroneous phrasing. Replace it with a more academic construction (“The results herein... “).
Line 221: A space is needed before “The“.
Line 221-222: It is very unclear what “the final dress“ refers to in this sentence. Clarifiy!
Lines 310-311: Too many brackets within brackets, rearrange the text.
Line 325: Wrong construction of the “if clause“, the grammatically correct version is “If consumers rethink the way...., they will... “.
Line 343: Wrong use of the gerund again, the correct version is “avoiding to consider“.
Line 347: Wrong word, replace “easy“ with „ease“.
The text contains several grammatical errors. Extensive editing of English language is required.
Author Response
The authors wish to thank you the reviewers for the suggestion provided; here below is the reply:
The Introduction has been structurally revised, as well as the Discussion section. Since it is a Case Report we consider sufficient the material presented. In addition, all the specific comments were addressed in the text.

Reviewer 4 Report
The paper should be published without major changes. The paper deals with an attractive subject of enhancing the sustainability and resilience in Small Scale Fisheries. The paper is well-written, clear, and gives robust results which support the conclusions. The quality of writing is good, with clearness and a good format of presentation. The scientific content is accurate, balanced, and interesting.
Please, I would like to comment on the following points. First, please complete all the literature (like #30 Knight…) accordingly and cross-check any edit errors. Moreover, the conclusions need more improvement to cover the main parts of your paper.
Author Response
The authors wish to thank you the reviewers for the suggestion provided; here below is the reply:
The literature format has been revised, as well as the Discussion and Conclusion sections.
Round 2
Reviewer 2 Report
Following revisions much of the manuscript is now greatly improved. In particular the introduction is much easier to follow and the aims and objectives of the work are clearer. I do note that the authors have stated that they use the EMFAF definition of SSF, so this excludes towed gears. I think this should be made clearer in the introduction rather than just referring to boats <12m in length.
I still feel there could be some improvements made to the methods section and the results section to make the co-design process in particular clearer. I don't think it is adequate enough for the authors to dismiss the comments from Reviewer 3 regarding the needs for extra details in these sections just because this manuscript represents a case report.
Specifically more detail could be added regarding pilot testing. Was the app tested by those who took part in the co-design process or others from outside of this group? Was it just tested for general ease of use/functionailty or was it trialed in real life scenarios? How many individuals were involved in pilot testing and from what sectors? How were results from pilot testing fed back into the design process?
The conclusion reflects on the capability of ICT tools to effectively contributing to promoting economic, social and environmental targets for SSF but I think the authors needs to be more explicit as how this co-design app will achieve this. Are there any results from testing that suggest this is the case? Or feedback from those involved in its development. I think the conclusions are weak in light of manuscript in its current state and some lacking of detail still in terms of the development/testing/roll out of the app.
Author Response
The authors thank the reviewers for their suggestions that lead to the improvement of the manuscript. In the introduction, the SSF definition from EMFAF has been clearly specified.
Additional details on the pilot testing were added to the Result section to let understand that the application is considered as the main result of the co-design process. It is now specified that only the stakeholders involved in the development of the app used it as a trial. The conclusion section was modified to better reflect the context of the manuscript.
Reviewer 3 Report
A final proofing for English should be done.
A final proofing for English should be done.
Author Response
The work was revised by a colleague fluent in English.